# Sociodynamics of Reinforcement Learning

**Yann Bouteiller**  *yann.bouteiller@polymtl.ca*
*Department of Computer Science*
*Polytechnique Montreal*

**Karthik Soma**
*Department of Computer Science*
*Polytechnique Montreal*

**Giovanni Beltrame**
*Department of Computer Science*
*Polytechnique Montreal*

**Reviewed on OpenReview:** *https://openreview.net/forum?id=Ro6Ylnx8se*

## Abstract

Reinforcement Learning (RL) has emerged as a core algorithmic paradigm explicitly driving innovation in a growing number of industrial applications, including large language models and quantitative finance. Furthermore, computational neuroscience has long found evidence of natural forms of RL in biological brains. Therefore, it is crucial for the study of social dynamics to develop a scientific understanding of how RL shapes population behaviors. We leverage the framework of Evolutionary Game Theory (EGT) to provide building blocks and insights toward this objective. We propose a methodology that enables simulating large populations of RL agents in simple game theoretic interaction models. More specifically, we derive fast and parallelizable implementations of two fundamental revision protocols from multi-agent RL - Policy Gradient (PG) and Opponent-Learning Awareness (LOLA) - tailored for population simulations of random pairwise interactions in stateless normal-form games. Our methodology enables us to simulate large populations of 200,000 independent co-learning agents, yielding compelling insights into how non-stationarity-aware learners affect social dynamics. In particular, we find that LOLA learners promote cooperation in the Stag Hunt model, delay cooperative outcomes in the Hawk-Dove model, and reduce strategy diversity in the Rock-Paper-Scissors model.

## 1 Introduction

Our universe is one of perpetual change, where countless agents co-exist and co-learn. From an individual agent's perspective, other learning agents are a fundamentally non-stationary part of the environment, especially when incentives are in conflict (Papoudakis et al., 2019). In the realm of Reinforcement Learning (RL), the study of this possibly adversarial non-stationarity is called Multi-Agent Reinforcement Learning (MARL) (Tan, 1993; Foerster, 2018; Zhang et al., 2021). Yet, MARL is largely constrained by the high complexity of multi-agent training. Its applications often resort to "self-play" (Silver et al., 2016; Berner et al., 2019), i.e., training a single neural network against one or few copies of itself, or to "centralized training" (Lowe et al., 2020; Yu et al., 2022), i.e., circumventing non-stationarity by using privileged global information. We are instead interested in decentralized, biologically plausible MARL processes that may help us understand the macroscopic social dynamics stemming from individual agents pursuing their own incentives in the real world. This paper is a step toward achieving this objective: we simulate large populations of RL agents pursuing individualistic incentives in classic stateless interaction models. At each population step, agents are paired by some assortment process (in our case, uniformly at random), interact according to their respective policies, and adapt their policies according to their respective payoffs and learning rules.

Our work brings advanced MARL to the scale of population game simulations. In particular, we study the population-scale effects of Opponent-Learning Awareness, an advanced MARL paradigm able to take advantage of non-stationary dynamics (Foerster et al., 2017). Our contribution is threefold:

- We propose a methodology for simulating RL-driven social dynamics in stateless interaction models,

- We derive tractable implementations of exact Policy Gradient (PG) and Learning with Opponent-Learning Awareness (LOLA) tailored for pairwise stateless population games,

- We analyze how learning shapes collective behavior in simulated populations of 200,000 agents.

## 2   Related work

Our paper constitutes a new step toward modeling the fast-paced macroscopic social dynamics that stem from individuals actively optimizing their own payoffs in real-world economies. We trace our line of work back to Macy & Flache (2002), who first substantiated the claim that the emphasis on the stochasticity-driven model of genetic evolution long used by evolutionary biologists to explain the evolution of cooperation (Axelrod & Hamilton, 1981) may need to shift to cognitive-driven dynamics. Concurrently, Sato & Crutchfield (2003) derived coupled replicator equations to describe how naive RL shapes social dynamics in a 2-player repeated Rock-Paper-Scissors interaction model. However, later attempts have remained largely constrained to imitation-driven dynamics (Sandholm, 2010; Xia et al., 2011). Presumably, this restriction can be explained by the high complexity of learning-driven population dynamics and by the strong heritage of replication-driven evolutionary genetics. The mathematical framework typically used to model population dynamics is Evolutionary Game Theory (EGT) (Smith & Price, 1973), which comes with a large body of imitation-based literature. Mertikopoulos & Sandholm (2018) have recently extended this framework by formalizing population dynamics (such as the Replicator Dynamic) under the more general class of Riemannian game dynamics. Further, Yang et al. (2020); Gu et al. (2021); Hu & Zhang (2025) used mean-field theory in a massively multi-agent RL scenario to reduce the environment dimensionality. Typically, in these approaches, naive learners approximate neighboring agents as one single, "mean-field" opponent. While mean-field methods are indeed relevant to studying large-scale population dynamics, they essentially transform $n$-agent MARL into pairwise MARL, thus reducing the environmental complexity from the point of view of individual agents. Alternatively, Christianos et al. (2021) reduced the complexity of MARL settings by selectively sharing parameters among agents, as a practical trade-off between self-play and independent learning. In contrast, our work explicitly tracks full populations of 200,000 independent RL policies by making simulation tractable in simple evolutionary game-theoretical interactions.

## 3   Background

This section briefly reviews the concepts from Evolutionary Game Theory (EGT) and Multi-Agent Reinforcement Learning (MARL) used in this paper. Throughout the paper, we interchangeably use vocabulary from EGT and MARL, whose correspondence is provided in the Appendix (Table 2).

### 3.1   Evolutionary Game Theory (EGT)

EGT (Smith & Price, 1973) models evolution as a series of random pairwise interactions, where interactions are typically simple bi-matrix games (i.e., 2-player multi-armed bandits). Agents are sampled from a large population to be randomly paired and evaluated against their drawn opponent. The outcome of this interaction is a *payoff* for each opponent, whose expectation is called the agent's *fitness* against the current population. An agent's fitness depends on both its policy and the current configuration of the population. The agent's policy is called its *type*, which is one of $n$ possible types. In gene-inspired *revision protocols*, agents with a greater fitness replicate and thus tend to "invade" the population, whereas agents with a lower fitness tend to go "extinct". In particular, EGT is interested in *evolutionarily stable equilibria*, which are configurations of the population where the different types are present in stable proportions under replication dynamics. In evolutionarily stable equilibria, the population configuration is robust to rare mutations, where

few agents randomly switch from one type to another. In this paper, we study more complex population dynamics generated by learning-based revision protocols (as opposed to revision protocols modeling genetic replication or social imitation) in three classic symmetric interaction models from game theory: *Stag Hunt*, *Hawk-Dove*, and *Rock-Paper-Scissors*. These interaction models are the following bi-matrices:

|          | Stag   | Hare   |
|----------|--------|--------|
| **Stag** | $s, s$ | $0, 1$ |
| **Hare** | $1, 0$ | $1, 1$ |

(a) Stag Hunt

|          | Hawk   | Dove   |
|----------|--------|--------|
| **Hawk** | $f, f$ | $2, 0$ |
| **Dove** | $0, 2$ | $1, 1$ |

(b) Hawk-Dove

|              | Rock    | Paper   | Scissors |
|--------------|---------|---------|----------|
| **Rock**     | $0, 0$  | $-1, 1$ | $1, -1$  |
| **Paper**    | $1, -1$ | $0, 0$  | $-1, 1$  |
| **Scissors** | $-1, 1$ | $1, -1$ | $0, 0$   |

(c) Rock-Paper-Scissors

where rows represent the action chosen by the ego agent (bold) with corresponding payoffs in first position, and columns represent the action chosen by the other agent with corresponding payoffs in second position. $s$ and $f$ are cost parameters, whose influence is explored in Section 5. The significance of these interaction models (or *games*) is further described in Appendix B for the reader unfamiliar with EGT.

A population whose individuals are distributed amongst $n$ types can be represented as a *population vector* $\underline{P} \in \mathbb{R}^n$ whose components $0 \leq \underline{p}_i \leq 1$ sum to 1 and represent the proportion of type $i$. Under the "imitation of the fittest" revision protocol (as well as several other revision protocols), large populations are known to follow a famous population dynamic over time ($t$), called the *Replicator Dynamic*:

$$\frac{d\underline{p}_i}{dt} = \underline{p}_i(\underline{v}_i(\underline{P}) - \bar{\underline{v}}(\underline{P})) \tag{1}$$

where $\underline{v}_i(\underline{P})$ is the fitness of type $i$ in the population, and $\bar{\underline{v}}(\underline{P})$ is the average fitness of all agents in the population. Denoting the vector of $\underline{v}_i$'s as $\underline{Q}$, the vector of all ones as $\mathbf{1}$ and the Hadamard product[1] as $\odot$, Equation 1 can be written in matrix form:

$$\frac{d}{dt}\underline{P} = \underline{P} \odot (\underline{Q} - \mathbf{1}\bar{\underline{v}}), \tag{2}$$

### 3.2 Multi-Agent Reinforcement Learning

We examine populations of persistent agents actively learning and optimizing their own policies in the pursuit of their own individualistic incentives. Following classic modeling of large populations in EGT, we model social dynamics as a series of random pairwise interactions. Therefore, we are principally interested in 2-agent learning rules. In this paper, we will be specifically looking at two important such learning rules: *policy gradient* (PG), also referred to as "naive learning" in the MARL literature, and *learning with opponent-learning awareness* (LOLA) in its true form (i.e., using all terms from the first-order Taylor expansion).

**Policy gradient** (PG) (Williams, 1992; Sutton et al., 1999) is a fundamental learning rule from single-agent RL. It follows the first-order gradient of the value function with respect to the ego agent's policy parameters. Let us consider a pair of agents. We denote the ego agent as agent 1, and the other agent as agent 2. Let us further denote their respective policies as $P_1$ and $P_2$, parameterized by vectors $\theta_1$ and $\theta_2$, of current values $v_1$ and $v_2$. The naive policy gradient is:

$$\nabla_{\theta_1} v_1(\theta_1, \theta_2) \tag{3}$$

The reason why following this gradient is considered naive in MARL is that this does not take into account the non-stationarity introduced by the learning process of agent 2.

**Learning with opponent learning awareness** (LOLA) (Foerster et al., 2017) is an improved version of PG that takes into account the learning process of the other agent. More precisely, LOLA models the learning process of agent 2 as if agent 2 were a naive learner, and differentiates through its learning step:

$$\nabla_{\theta_1} v_1(\theta_1, \theta_2 + \Delta\theta_2) \tag{4}$$

---

[1]The matrix product takes precedence over the Hadamard product in all our notations.

where $\Delta\theta_2 = \eta\nabla_{\theta_2}v_2(\theta_1, \theta_2)$ is the naive learning step of agent 2, $\eta$ being its learning rate. LOLA approximates this gradient with the following first-order Taylor expansion (Taylor, 1717):

$$\nabla_{\theta_1}v_1(\theta_1, \theta_2 + \Delta\theta_2) \approx \nabla_{\theta_1}(v_1 + (\Delta\theta_2)^\top\nabla_{\theta_2}v_1)$$
$$= \underbrace{\nabla_{\theta_1}v_1}_{\text{PG}} + \underbrace{\eta(\nabla_{\theta_1}\nabla_{\theta_2}v_1)^\top\nabla_{\theta_2}v_2 + \eta(\nabla_{\theta_1}\nabla_{\theta_2}v_2)^\top\nabla_{\theta_2}v_1}_{\text{opponent-learning awareness}} \quad (5)$$

Differentiating through the learning step of the opponent has an important advantage in our discussion: it is a naturally plausible way of predicting non-stationarity (assuming we maintain an internal model of others) in order to adapt beforehand and actively steer this non-stationarity toward our own incentives.

### 3.3 Population-policy equivalence

In a population of agents playing only pure strategies, uniformly sampling agents is equivalent to sampling actions from the abstract stochastic policy defined by the probability vector $P = \underline{P}$. Thus, Equation 2 can be viewed as a learning process, albeit at the population level, where the evolving population of non-learning agents $\underline{P}$ is itself a self-play learning agent (Bloembergen et al., 2015).

## 4 Methods

To simulate how learning affects societies, we adopt the philosophy of EGT. Namely, we consider large populations of independent learning agents, which are paired randomly at each evolution iteration and interact in normal-form matrix games. Each agent has its own learning rule (i.e., either of the two presented in Section 3.2) that it applies to its own policy after each pairwise interaction. Whereas MARL usually thinks about these rules in the context of persistent interactions between fixed pairs of agents, in the context of population games, they instead get applied after single interactions between random pairs of agents. In other words, from the perspective of a learning agent, the opponent it samples at each step is a stochastic sample of the population and, for LOLA, of the current direction of its non-stationary dynamics.

### 4.1 Policy architecture

From an RL perspective, the normal-form matrix games presented in Section 3.1 are 2-agent multi-armed bandits. As this is a common assumption in multi-armed bandits (Sutton, 2018), we consider the policy architecture parameterized by the preference vector $\theta \in \mathbb{R}^n$, where $n$ is the number of actions, projected to the probability simplex by a simple softmax function $\sigma$, which yields the probability vector $P \in \mathbb{R}^n$ of the policy selecting each of the $n$ available actions:

$$P = \sigma(\theta) \quad (6)$$

This policy architecture has a useful property for our derivations: its gradient has a symmetric analytical form, which is

$$\nabla_\theta P = (\nabla_\theta P)^\top = \text{diag}(P) - PP^\top \quad (7)$$

### 4.2 Analytical Policy Gradient

Let us consider a symmetric normal-form game with $n$ actions, played by a pair of agents denoted as agents 1 and 2. Since the game is symmetric, we can represent its bi-matrix as a single matrix $A \in \mathbb{R}^{n \times n}$, valid from the perspective of both agents[2]. Let us further assume that the policies of both agents are parameterized by $\theta_{1,2} \in \mathbb{R}^n$, with the simple policy architecture described in Equation 6:

$$P_{1,2} = \sigma(\theta_{1,2}) \quad (8)$$

The value functions of both agents are:

$$v_1 = P_1^\top A P_2 \ ; \ v_2 = P_2^\top A P_1 \quad (9)$$

---

[2]$A$ is formed by the first entries of the corresponding bi-matrix in Section 3.1.

Thus, the "naive" Policy Gradient of agent 1's value with respect to agent 1's parameters is:

$$\nabla_{\theta_1} v_1 = P_1 \odot (Q_1 - \mathbf{1}v_1)$$

where $Q_1 \in \mathbb{R}^n$ is the vector of action-values of the $n$ available actions (derivation in Appendix D).

As a side note, this draws an interesting parallel with Equation 2: the PG update on the parameter vector $\theta_1$ is the same as the Replicator update on the probability vector $P_1 = \sigma(\theta_1)$. In other words, each individual PG agent can itself be seen as an evolving population of abstract non-learning agents playing pure strategies, similarly to the equivalence noted in Bloembergen et al. (2015).

This formulation yields the following analytical PG formulation for symmetric normal-form games:

$$\nabla_{\theta_1} v_1 = P_1 \odot (I - \mathbf{1}P_1^\top)AP_2 \tag{10}$$
$$\nabla_{\theta_2} v_2 = P_2 \odot (I - \mathbf{1}P_2^\top)AP_1 \tag{11}$$

which only involves simple matrix operations, and therefore is a fast, easily parallelizable implementation.

### 4.3 Analytical LOLA

Similarly, we derive the following analytical formulation of the LOLA gradient in the symmetric normal-form game defined by matrix $A$ (the derivation is provided in Appendices E and F):

$$\begin{aligned}
\nabla_{\theta_1} v_1(\theta_1, \theta_2 + \Delta\theta_2) \approx\ & P_1 \odot X_1 AP_2 \\
&+ \eta(T^\top \odot X_1 AX_2^\top)(P_2 \odot X_2 AP_1) \\
&+ \eta(T^\top \odot X_1 A^\top X_2^\top)(P_2 \odot X_2 A^\top P_1)
\end{aligned} \tag{12}$$

where $X_1 := I - \mathbf{1}P_1^\top$, $X_2 := I - \mathbf{1}P_2^\top$, and $T := P_2 P_1^\top$. As for Policy Gradient, this formulation only involves simple matrix operations and is thus straightforward to parallelize.

### 4.4 Batched pairwise bandits

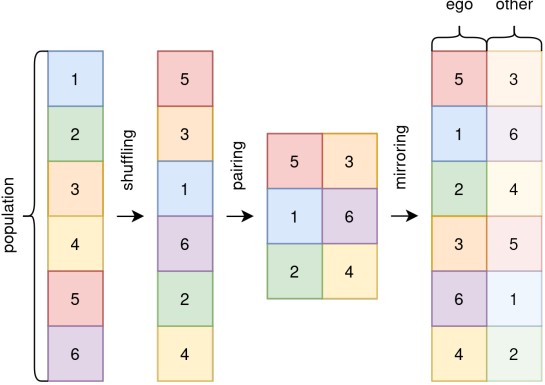

Figure 1: Pairing and batching. The population is shuffled and split to produce a random pairwise matching. The resulting tensor is then mirrored such that the entire population step can be performed in parallel.

Equations 10 and 12 are fast, backprop-free implementations of exact PG and exact LOLA for symmetric normal-form games. Still, it would be prohibitively slow to apply these updates iteratively on single agent pairs in large-scale simulations. At each *population step*, all agents go through their revision protocol once. To make this process scalable, we batch learning updates across the entire population. More precisely, at each population step, we shuffle the entire population and randomly pair all agents two-by-two. Since all interactions are pairwise, this enables batching updates across agent pairs. To optimize the process even further, we mirror all pairs to perform the entire population update in one single batched operation.

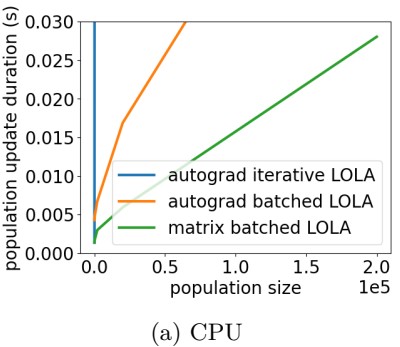

(a) CPU

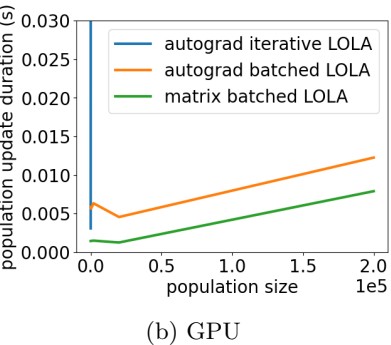

(b) GPU

Figure 2: Duration of a full population step (lower is better). We benchmark different implementations of analytical LOLA. Both batched approaches implement Section 4.4, and the matrix approach implements Section 4.3. The matrix batched approach's efficiency enables full simulations of very large populations.

This batching procedure is illustrated in Figure 1. We found batching populations in this manner to be extremely efficient. Combining this procedure with the analytical PG and LOLA implementations described in Section 4, we are able to simulate populations of 200,000 learning agents for thousands of population steps in a matter of seconds on a consumer-grade GPU[3]. To ensure reproducibility and foster future work, we open-source our our code at `https://github.com/MISTLab/RL-societies`.

## 5 Experiments

We use the normal-form matrices presented in Section 3.1 as our three explored interactions: *Stag hunt* (SH), *Hawk-Dove* (HD) and *Rock-Paper-Scissors* (RPS). Each individual agent has its own persistent learning rule: either PG (gradient ascent on Equation 10) or LOLA (gradient ascent on Equation 12). We use unit learning rates in all experiments except when stated otherwise.

### 5.1 Scalability

Figure 2 reports the computational performance of our approach ("matrix batched LOLA"), compared to two baselines. The "autograd iterative LOLA" baseline reproduces how LOLA updates are usually performed in classical MARL scenarios: using PyTorch's autograd to compute the LOLA gradient, and updating the policies of all agents iteratively. This baseline is clearly not a viable implementation for large population simulations and is only provided for illustration. On the other hand, our "autograd batched LOLA" baseline is of more interest for future work. While the "matrix batched" approach is significantly faster, it is limited to single-shot multi-armed bandits[4]. In particular, the "matrix batched" approach does not allow episodic interactions. Therefore, we have implemented the batched approach described in Section 4.4 along with autograd, which yields a potentially more general implementation. Since our matrix-based implementation is the fastest for normal-form games, we use it in the remainder of the paper.

### 5.2 Empirical results

The methodology proposed in Section 4 is limited to simple, single-shot bandit interactions between random pairs of learning agents. Remember how, under the population-policy equivalence described in Section 3.3, a population of pure-strategy agents can equivalently be seen as a stochastic policy over types. In our setting, agents have full stochastic policies assigning non-zero probabilities to all available actions, but they are still simple stateless multi-armed bandits. Uniformly sampling a random pair of agents from such a population and then sampling from their policies is equivalent in expectation to sampling two actions from the average policy of the entire population. In other words, we expect our modeled population dynamics to show some

---

[3]All experiments in this paper are conducted with an i7-12700H CPU, an RTX 3080 Ti GPU, and 64G of RAM.
[4]Section 4 is possible because the value function has a straightforward formulation in 2-agent multi-armed bandits.

resemblance to self-play over the population's average policy (represented in Figure 3). In Figure 4, we report the results of our actual population experiments in Stag-Hunt, Hawk-Dove and Rock-Paper-Scissors. Due to the large number of simulated agents, the reported population dynamics are near-deterministic.

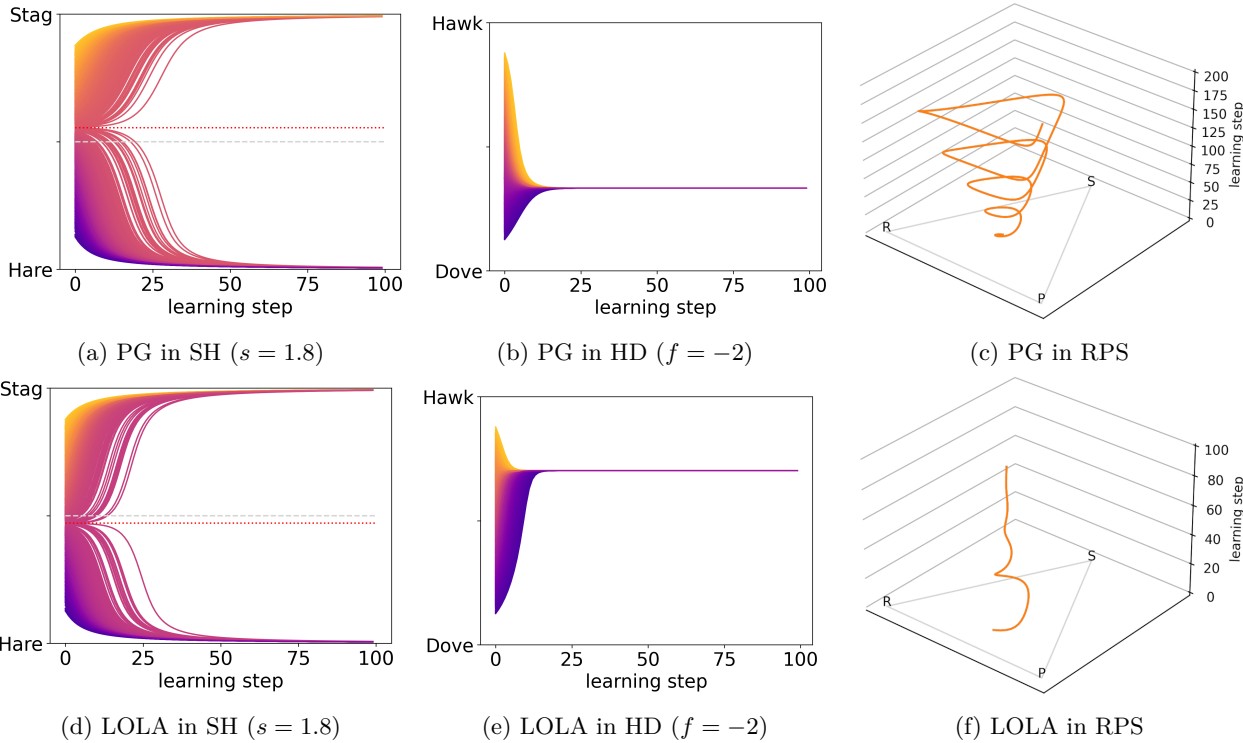

Figure 3: Evolution of a self-play agent following Policy Gradient or LOLA (rows) in the classic games of Stag Hunt, Hawk-Dove and Rock-Paper-Scissors (columns). In Stag Hunt and Hawk-Dove, colors mark the agent's initial policy (yellow for deterministic Stag/Hawk and blue for deterministic Hare/Dove). In Stag Hunt, gray dashes mark the uniform policy and red dots highlight the critical initial policy.

**Stag Hunt.** Figures 3a and 3d show how a single self-play agent learns against itself in Stag Hunt, via naive Policy Gradient and LOLA, respectively. The vertical axis represents the agent's policy and the horizontal axis represents time expressed in learning steps. Policies are color-coded by their initial configuration: yellow represents policies that start close to the deterministic Stag policy and purple represents those that start close to the deterministic Hare policy. Notably, PG tends to converge to the individualistic Nash equilibrium (i.e., deterministic Hare) for most initial configurations, whereas LOLA tends to converge to the pro-social Nash equilibrium (i.e., deterministic Stag). Notice that the forks (red dots) are on either sides of the uniform random policy (middle tick), which is important because we will initialize all our population experiments with a Gaussian distribution around this neutral policy. We expect the final population dynamic to follow a similar pattern. Figures 4a and 4d report the results of our first population game simulation, featuring 200,000 learning agents in the Stag Hunt interaction model. Dark shades of blue represent high concentrations of agents. lighter shades represent lower concentrations. A *Population steps* correspond to one learning step performed per agent in the population. In Figure 4a, the population is exclusively formed of naive learners, and quickly converges to the individualistic policy, as predicted by Figure 3a. In Figure 4d, the population is formed of only LOLA agents. Contrary to the naive population, a population of non-stationarity aware learners such as LOLA evolves to unanimously adopt the superior pro-social equilibrium (i.e., deterministic Stag). This effect is modulated by the payoff of the pro-social strategy, as shown in Figure 5a. In Appendix G, we further show that, when enough opponent-learning aware learners are present in a mixed population, they become able to pull naive learners toward the pro-social strategy.

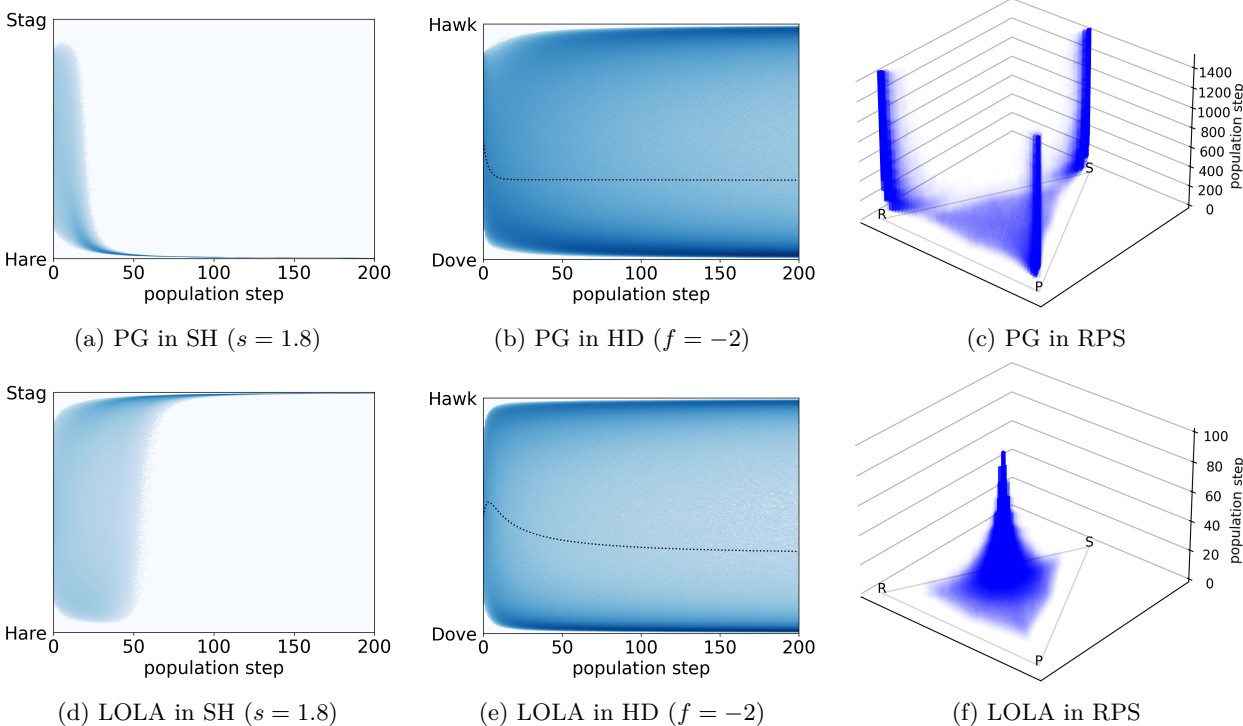

Figure 4: Populations of 200,000 RL agents in the Stag Hunt, Hawk-Dove and Rock-Paper-Scissors interaction models (columns), individually learning via Policy Gradient and LOLA (rows). Each agent is a stochastic policy, represented as linear coordinates between pure strategies. Populations are randomly initialized with a Gaussian around the uniform policy. Dark shades of blue indicate high concentrations of agents, and population steps correspond to one learning step performed per agent. In Hawk-Dove, black dots indicate the average policy over the entire population.

**Hawk-Dove**. Figures 3b and 3e show that, in Hawk-Dove, self-play converges to definite policies regardless of where training starts from. Naive learning (Figure 3b) converges to the mixed Nash equilibrium. However, LOLA (Figure 3e) converges to another, inferior policy, where it selects Hawk 70% of the time (which yields a smaller payoff for both players, and is not a Nash equilibrium). A similar behavior has been described as "arrogance" in Letcher et al. (2018), where both LOLA learners make wrong assumptions about the response of their opponent and thus pull away from the equilibrium.

From these observations, one could imagine that all learning agents in the population would converge to these policies, similar to what we observed for SH, but this is not at all what happens in practice. In HD, whether naive learning (Figure 4b) or LOLA (Figure 4e) is used as the learning rule of the entire population, it evolves into a mix of Hawks and Doves, most of them with close-to-deterministic policies, and in both cases with an average population policy that corresponds to the mixed Nash equilibrium.

Notice that the convergence to deterministic strategies is however much slower than what we observed for SH, and it is in fact not clear whether this will eventually happen entirely, even after 10,000 steps (Appendix H). Nonetheless, what can be observed from Figures 4b and 8a is that LOLA learners converge faster to deterministic policies during early steps (shades of blue) but the population takes longer to stabilize (dotted black). In additional experiments, we mixed LOLA and PG learners to see whether LOLA learners would be more inclined toward the Hawk strategy (as suggested by Figure 3e). However, these experiments invalidated this hypothesis: half LOLA learners and half PG learners were present amongst both final sub-populations of Hawk-inclined and Dove-inclined individuals.

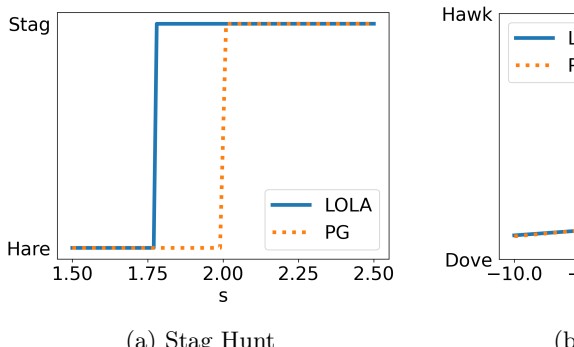

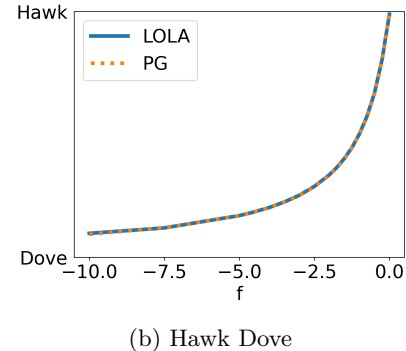

(a) Stag Hunt                    (b) Hawk Dove

Figure 5: Final average policy over the population, depending on cost values. In Stag Hunt, populations of LOLA agents deterministically cooperate more often compared to populations of PG agents, whereas in Hawk-Dove the final average policy depends only on $f$.

**Rock-Paper-Scissors.** Our final population experiment explores the 3-action Rock-Paper-Scissors interaction model, used in EGT to explain the coexistence of competitively unbalanced species (Allesina & Levine, 2011). We show that, while populations of naive learners agree with this explanation of sustainable diversity, populations of opponent-learning aware learners yield the opposite result. Similarly to previous sections, Figures 3c and 3f report how 2-agent self-play behaves for both PG and LOLA. Only one policy is displayed for readability (other initial conditions yield similar effects). The triangle on the bottom of each plot represents the policy, and the vertical axis represents the number of learning steps. It can be seen that PG slowly spirals outward from the mixed Nash equilibrium (due to performing straight policy updates with a non-zero learning rate following a circular vector field), whereas LOLA quickly spirals inward until it reaches the mixed Nash equilibrium. Figures 4c and 4f present the results of our population simulations in the RPS interaction. Color-coding follows the same principle as our previous population plots, with the bottom triangle being the policy simplex, and population steps being the vertical dimension. Similarly to Figure 4b, Figure 4c shows that populations of naive learners evolve into 3 equally distributed groups of close-to-deterministic agents always playing Rock, Paper and Scissors respectively. The reason why this happens is of interest, and is more clearly understood from Figure 10 in Appendix I. In short, this dynamic results from the loss of plasticity introduced by the softmax function[5]. Initially, naive learners are erratically moving around as they encounter all types of strategies. However, after some time, agents get "trapped" near the border of the policy simplex, where gradients toward the opposite action are near-zero. After a long time, three groups of near-deterministic agents emerge, and a small number of them continuously escape toward the strategy that counters the majority group, which eventually creates a new majority, and so on, yielding a cyclic social dynamic. In other words, diversity emerges from populations of naive learners in the RPS model. On the other hand, Figure 4f tells the opposite story about populations of LOLA agents, which instead quickly and unanimously converge to the mixed Nash equilibrium.

## 5.3 Influence of learning rates

When ascending the gradients described in Sections 4.2 and 4.3, RL agents must select learning rates to scale policy updates. PG agents require a single, first-order learning rate $\alpha$ and LOLA agents require an additional, second-order learning rate $\eta$. In our work, both learning rates are constant and shared across the population. So far, we have only considered unit learning rates $\alpha = \eta = 1$. However, a puzzling effect emerges in the Stag Hunt interaction model when varying $\alpha$ and $\eta$ independently. Figures 6d and 6b report the strategy selected on average by populations of LOLA agents in Stag Hunt and Hawk Dove, depending on $\alpha$ and $\eta$. Similar to Figure 4, populations are initialized with a Gaussian around the uniformly random policy. Figures 6a and 6c report the policy to which a self-play LOLA agent converges under the same conditions, when initialized at the uniformly random policy. Intuitively, in LOLA, the first-order learning rate $\alpha$ scales the ego policy update whereas the second-order learning rate $\eta$ scales the simulated policy

---

[5]A model similar to the "cost of motion" described by Mertikopoulos & Sandholm (2018).

update of the opponent. When $\eta$ is close to 0, opponent-learning awareness vanishes and LOLA becomes PG. This is consistent with what we observe for Hawk-Dove: Figure 6c shows the gradual effect of arrogance as $\eta$ increases, while Figure 6b shows that this effect vanishes in large populations, as we previously observed in Figure 4e. However, Figures 6a and 6d unveil a much more surprising phenomenon: whereas only $\eta$ has a significant effect on the final policy of a self-play agent in Stag Hunt, the first-order learning rate $\alpha$ has a significant combined effect with $\eta$ in large populations. We do not yet fully understand this phenomenon, and therefore we leave its explanation as an avenue for future work.

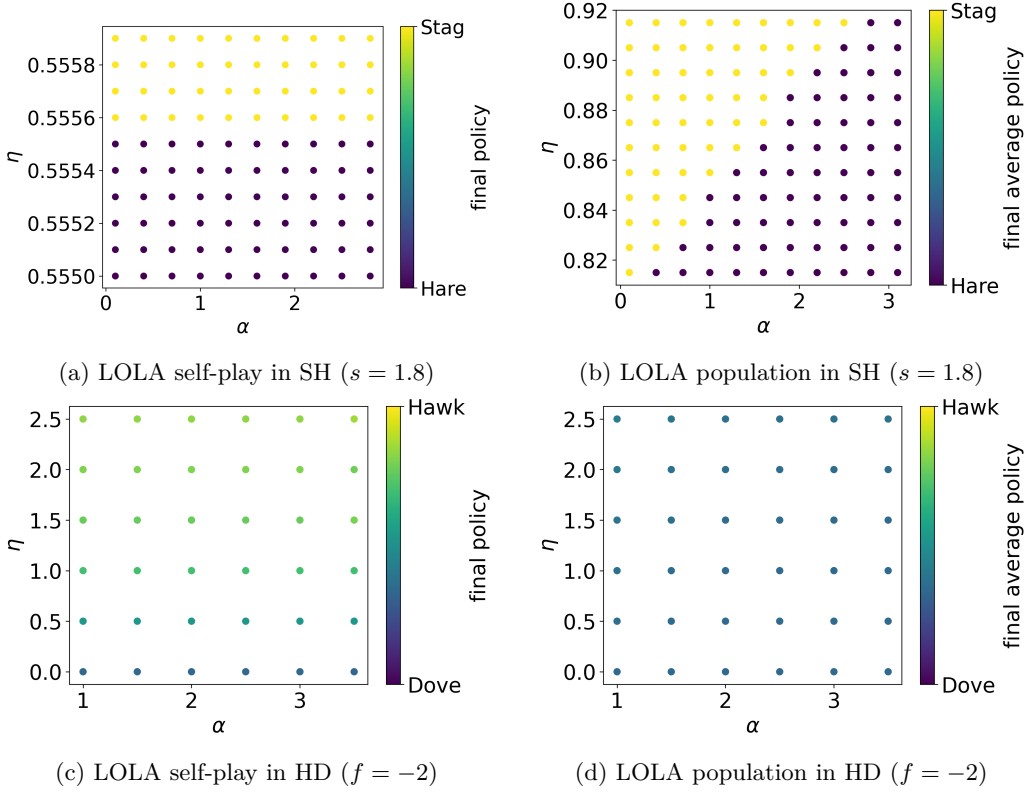

(a) LOLA self-play in SH ($s = 1.8$)

(b) LOLA population in SH ($s = 1.8$)

(c) LOLA self-play in HD ($f = -2$)

(d) LOLA population in HD ($f = -2$)

Figure 6: Influence of the first and second-order learning rates $\alpha$ and $\eta$ in Stag Hunt and Hawk-Dove. Surprisingly, in Stag Hunt, $\alpha$ influences the strategy adopted by populations of LOLA agents, even though this influence does not exist in self-play. In Hawk-Dove, the average strategy adopted within the population is unaffected by learning rates, unlike that of self-play which is directly affected by $\eta$ due to "arrogance".

## 5.4 Limitations and future work

**Random versus structured assortment.** Our simulations can mislead the reader into concluding that the mean policy averaged over the entire population always converges near a Nash equilibrium of the game, even when the learning rule itself does not converge to this equilibrium in the conventional 2-agent MARL setting (see Figures 3e and 3c). This property is however merely a consequence of the uniform random opponent matching scheme that we chose to implement in this paper. For instance, let us consider an extreme opposite scheme, where all pairs would instead interact persistently. All individual pairs would then converge to the pure Nash equilibrium in the Hawk-Dove game (that is, exactly one deterministic Hawk and one deterministic Dove per pair): this would average to a uniform random policy, as opposed to Figures 4b and 4e (dotted black lines). In reality, partner selection is more structured (Anastassacos et al., 2020) and can lead to different outcomes. Extending our study to structured assortment is an avenue for future work.
**Stateless environments.** Our proposed approach is limited to symmetric normal-form matrix games (i.e., stateless multi-armed bandit interactions). Exploring stateful episodic interactions (like the Iterated Prisoner's Dilemma) through more compute-intensive approaches is another clear avenue for future work.

# 6 Conclusion

We have presented a methodology enabling large-scale population simulations of independent learning agents, for both naive (Policy Gradient) and advanced non-stationarity-aware (LOLA) learning rules. We have demonstrated the scalability of our approach by performing very-large-scale simulations of 200,000 independent learning agents, interacting in the classic games of Stag Hunt, Hawk-Dove and Rock-Paper-Scissors. Our work essentially explores the effect of Multi-Agent Reinforcement Learning on classic evolutionary game theoretical models of social dynamics, and demonstrates compelling dynamics originating from both naive and non-stationarity-aware learners. For instance, depending on the nature of the interaction, opponent-aware learners can foster cooperation, delay cooperative outcomes, or inhibit diversity in the population.

**Broader Impact Statement**

This work paves the way toward understanding and possibly exploiting the macroscopic effects of individual learning in contexts such as biology, sociology, economy and finance.

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

## A    Vocabulary

| RL | EGT |
|---|---|
| return | payoff |
| value | fitness |
| policy | type |
| action | pure strategy |
| learning rule | revision protocol |

Table 2: Correspondence of the vocabulary from RL and EGT

## B    Games

|  | Stag | Hare |
|---|---|---|
| **Stag** | $s, s$ | $0, 1$ |
| **Hare** | $1, 0$ | $1, 1$ |

(a) Stag Hunt

|  | Hawk | Dove |
|---|---|---|
| **Hawk** | $f, f$ | $2, 0$ |
| **Dove** | $0, 2$ | $1, 1$ |

(b) Hawk-Dove

|  | Rock | Paper | Scissors |
|---|---|---|---|
| **Rock** | $0, 0$ | $-1, 1$ | $1, -1$ |
| **Paper** | $1, -1$ | $0, 0$ | $-1, 1$ |
| **Scissors** | $-1, 1$ | $1, -1$ | $0, 0$ |

(c) Rock-Paper-Scissors

**Stag hunt** (SH) is a 2-action game modeling a cooperative dilemma. In this game, agents need to hunt for food and choose to either go for a Stag, or go for a Hare. Hunting a Hare is easy: any agent choosing this option successfully receives a payoff of 1. Hunting a Stag is harder: both agents need to cooperate, otherwise the agent choosing to go for a Stag fails to catch anything and receives a payoff of 0. However, if both agents cooperate, they succeed and each receives a payoff of $s > 1$, which is better than going for Hares. The game of Stag hunt has two distinct pure strategy Nash equilibria[6]: (1) both agents always playing Stag, and (2) both agents always playing Hare.

**Hawk-Dove** (HD) is a 2-action game modeling conflict over shareable resources. Agents either choose to act as a "Hawk" or as a "Dove". When a Dove encounters another Dove, they share the available food (each receives a payoff of 1). When a Dove encounters a Hawk, it yields and gets no food (payoff of 0) while the Hawk gets all of it (payoff of 2). But when a Hawk encounters another Hawk, they fight and both get injured (payoff of $f < 0$). The game of Hawk-Dove has two pure strategy Nash equilibria: (1) Hawk-Dove and (2) Dove-Hawk. But note that in these equilibria, Doves have a smaller payoff than Hawks. In the context of evolutionary genetics, this means that when Doves encounter almost only Hawks, they move toward extinction as Hawks invade. However, when Hawks encounter almost always Hawks, their expected payoff is even less than Doves encountering Hawks, and thus Hawks move toward extinction as Doves invade. In other words, there are population configurations in which it is not more relevant to be a Hawk than a Dove in terms of fitness, and replication dynamics naturally drive the population there.

**Rock-Paper-Scissors** (RPS) is a 3-action zero-sum[7] game. It illustrates more complex situations where there are cycles in the preferences over actions. Scissors beats Paper, Paper beats Rock, and Rock beats Scissors. RPS has a single mixed-Nash equilibrium, where both players choose their actions uniformly at random. Similarly to the HD game, all populations whose average behavior is this equilibrium are neutrally stable under replication dynamics and drifting due to random mutations.

## C    Parallels between EGT and single-agent RL

The population-policy equivalence described by Bloembergen et al. (2015) yields interesting parallels between single-agent Reinforcement Learning and the Replicator Dynamic. In particular, the resemblance of Policy Gradient with the Replicator Dynamic noted in Section 4.2 was further studied by Hennes et al. (2020). From

---

[6]2-player equilibria where each agent always selects the same action.
[7]The sum of the two agents' payoffs is always 0.

this observation, they derived a single-agent algorithm that bypasses the loss of plasticity introduced by the softmax architecture of Equation 6. However, beyond the fact that their line of work uses concepts from RL and EGT, it is unrelated to ours and we cite it here to clear a confusion made by early reviewers of our work: whereas Hennes et al. (2020) are interested in finding high-performance single-agent RL algorithms, we are interested in characterizing the population dynamics that stem from aggregated MARL.

## D Policy gradient

We derive an analytical formulation of the PG update in symmetric normal-form games:

$$
\begin{aligned}
\nabla_{\theta_1} v_1 &= \nabla_{\theta_1} P_1^\top A P_2 \\
&= (\operatorname{diag}(P_1) - P_1 P_1^\top) A P_2 \\
&= \operatorname{diag}(P_1) A P_2 - P_1 v_1 \\
&= P_1 \odot (A P_2 - v_1 \mathbf{1}) \\
&= P_1 \odot (A P_2 - \mathbf{1} P_1^\top A P_2) \\
&= P_1 \odot (Q_1 - \mathbf{1} v_1)
\end{aligned}
$$

## E LOLA

We now derive an analytical formulation of the LOLA update in symmetric normal-form games, similar to what we found for PG in Section 4.2. We are missing three terms from Equation 5:

- $\nabla_{\theta_2} v_1$

- $\nabla_{\theta_1} \nabla_{\theta_2} v_1$

- $\nabla_{\theta_1} \nabla_{\theta_2} v_2$

To compute the first term, we note that $v_1 = P_1^\top A P_2$ is a scalar and thus can also be written $v_1 = P_2^\top A^\top P_1$. We can then compute this term similarly to PG:

$$
\begin{aligned}
\nabla_{\theta_2} v_1 &= \nabla_{\theta_2} P_2^\top A^\top P_1 \\
&= (\operatorname{diag}(P_2) - P_2 P_2^\top) A^\top P_1 \\
&= P_2 \odot (A^\top P_1 - v_1 \mathbf{1}) \\
&= P_2 \odot (A^\top P_1 - \mathbf{1} P_2^\top A^\top P_1) \\
&= P_2 \odot (I - \mathbf{1} P_2^\top) A^\top P_1
\end{aligned}
\tag{13}
$$

Computing the two remaining terms is also possible.
Let us start with $\nabla_{\theta_1} \nabla_{\theta_2} v_2$:

$$
\nabla_{\theta_1} \nabla_{\theta_2} v_2 = \nabla_{\theta_1} \nabla_{\theta_2} P_2^\top A P_1
\tag{14}
$$
$$
= \nabla_{\theta_1} (\operatorname{diag}(P_2) - P_2 P_2^\top) A P_1
$$
$$
= (\operatorname{diag}(P_2) - P_2 P_2^\top) A (\operatorname{diag}(P_1) - P_1 P_1^\top)
\tag{15}
$$

While it would already possible to implement this formulation, we further derive a more efficient implementation in Appendix F:

$$
\nabla_{\theta_1} \nabla_{\theta_2} v_2 = T \odot (I - \mathbf{1} P_2^\top) A (I - P_1 \mathbf{1}^\top)
\tag{16}
$$

where $T := P_2 P_1^\top$.

Computing $\nabla_{\theta_1}\nabla_{\theta_2}v_1$ is fairly straightforward:

$$
\begin{aligned}
\nabla_{\theta_1}\nabla_{\theta_2}v_1 &= \nabla_{\theta_1}\nabla_{\theta_2}P_2^\top A^\top P_1 \\
&= \nabla_{\theta_1}\nabla_{\theta_2}P_2^\top B P_1 && (B := A^\top) \\
&= T \odot (I - \mathbf{1}P_2^\top)B(I - P_1\mathbf{1}^\top) && \text{(c.f. 14,16)} && (17) \\
&= T \odot (I - \mathbf{1}P_2^\top)A^\top(I - P_1\mathbf{1}^\top) && && (18)
\end{aligned}
$$

Substituting Equations 10, 11, 13, 16 and 18 in Equation 5 yields the following analytical formulation of the LOLA gradient in the symmetric normal-form game defined by matrix $A$:

$$
\begin{aligned}
\nabla_{\theta_1}v_1(\theta_1, \theta_2 + \Delta\theta_2) \approx\ & P_1 \odot X_1 A P_2 \\
& + \eta(T^\top \odot X_1 A X_2^\top)(P_2 \odot X_2 A P_1) \\
& + \eta(T^\top \odot X_1 A^\top X_2^\top)(P_2 \odot X_2 A^\top P_1)
\end{aligned}
\tag{19}
$$

where $X_1 := I - \mathbf{1}P_1^\top$, $X_2 := I - \mathbf{1}P_2^\top$, and $T := P_2 P_1^\top$.

# F  Second-order policy gradients

In this Section, we show that:

$$
\nabla_{\theta_1}\nabla_{\theta_2}v_2 = T \odot (I - \mathbf{1}P_2^\top)A(I - P_1\mathbf{1}^\top)
$$

where $T := P_2 P_1^\top$ is agent 2's transition matrix.

*Proof.*

$$
\begin{aligned}
\nabla_{\theta_1}\nabla_{\theta_2}v_2 &= \operatorname{diag}(P_2) - P_2 P_2^\top)A(\operatorname{diag}(P_1) - P_1 P_1^\top) \\
&= \operatorname{diag}(P_2)A\operatorname{diag}(P_1) - \operatorname{diag}(P_2)A P_1 P_1^\top - P_2 P_2^\top A\operatorname{diag}(P_1) + P_2 P_2^\top A P_1 P_1^\top
\end{aligned}
\tag{20}
$$

Note that, for $X, Y \in \mathbb{R}^n$:

$$
XY^\top = X\mathbf{1}^\top \odot \mathbf{1}Y^\top
$$

since:

$$
\begin{pmatrix}
x_1 y_1 & x_1 y_2 & \ldots & x_1 y_n \\
x_2 y_1 & x_2 y_2 & \ldots & x_2 y_n \\
\vdots & \vdots & & \vdots \\
x_n y_1 & x_n y_2 & \ldots & x_n y_n
\end{pmatrix}
=
\begin{pmatrix}
x_1 & x_1 & \ldots & x_1 \\
x_2 & x_2 & \ldots & x_2 \\
\vdots & \vdots & & \vdots \\
x_n & x_n & \ldots & x_n
\end{pmatrix}
\odot
\begin{pmatrix}
y_1 & y_2 & \ldots & y_n \\
y_1 & y_2 & \ldots & y_n \\
\vdots & \vdots & & \vdots \\
y_1 & y_2 & \ldots & y_n
\end{pmatrix}
$$

Also, for $M \in \mathbb{R}^{n,n}$ note that:

$$
\operatorname{diag}(X)M = X\mathbf{1}^\top \odot M
$$

since:

$$
\begin{pmatrix}
x_1 m_{1,1} & x_1 m_{1,2} & \ldots & x_1 m_{1,n} \\
x_2 m_{2,1} & x_2 m_{2,2} & \ldots & x_2 m_{2,n} \\
\vdots & \vdots & & \vdots \\
x_n m_{n,1} & x_n m_{n,2} & \ldots & x_n m_{n,n}
\end{pmatrix}
=
\begin{pmatrix}
x_1 & x_1 & \ldots & x_1 \\
x_2 & x_2 & \ldots & x_2 \\
\vdots & \vdots & & \vdots \\
x_n & x_n & \ldots & x_n
\end{pmatrix}
\odot
\begin{pmatrix}
m_{1,1} & m_{1,2} & \ldots & m_{1,n} \\
m_{2,1} & m_{2,2} & \ldots & m_{2,n} \\
\vdots & \vdots & & \vdots \\
m_{n,1} & m_{n,2} & \ldots & m_{n,n}
\end{pmatrix}
$$

And similarly:

$$
M\operatorname{diag}(X) = (\operatorname{diag}(X)M^\top)^\top = (X\mathbf{1}^\top \odot M^\top)^\top = M \odot \mathbf{1}X^\top
$$

So, taking a closer look at each term in Equation 20:

$$\text{diag}(P_2)A\,\text{diag}(P_1) = T \odot A$$

$$
\begin{aligned}
\text{diag}(P_2)AP_1P_1^\top &= P_2\mathbf{1}^\top \odot AP_1P_1^\top \\
&= P_2\mathbf{1}^\top \odot AP_1\mathbf{1}^\top \odot \mathbf{1}P_1^\top \\
&= P_2\mathbf{1}^\top \odot \mathbf{1}P_1^\top \odot AP_1\mathbf{1}^\top \\
&= T \odot AP_1\mathbf{1}^\top
\end{aligned}
$$

$$
\begin{aligned}
P_2P_2^\top A\,\text{diag}(P_1) &= P_2P_2^\top A \odot \mathbf{1}P_1^\top \\
&= P_2\mathbf{1}^\top \odot \mathbf{1}P_2^\top A \odot \mathbf{1}P_1^\top \\
&= P_2\mathbf{1}^\top \odot \mathbf{1}P_1^\top \odot \mathbf{1}P_2^\top A \\
&= T \odot \mathbf{1}P_2^\top A
\end{aligned}
$$

$$P_2P_2^\top AP_1P_1^\top = P_2v_2P_1^\top = T \odot v_2\mathbf{1}\mathbf{1}^\top$$

This enables writing the LOLA second-order gradient as:

$$\nabla_{\theta_1}\nabla_{\theta_2}v_2 = T \odot (A - AP_1\mathbf{1}^\top - \mathbf{1}P_2^\top A + v_2\mathbf{1}\mathbf{1}^\top)$$

The term between parentheses can be factorized:

$$
\begin{aligned}
A - AP_1\mathbf{1}^\top - \mathbf{1}P_2^\top A + v_2\mathbf{1}\mathbf{1}^\top &= A - AP_1\mathbf{1}^\top - \mathbf{1}P_2^\top A + \mathbf{1}v_2\mathbf{1}^\top \\
&= A - AP_1\mathbf{1}^\top - \mathbf{1}P_2^\top A + \mathbf{1}P_2^\top AP_1\mathbf{1}^\top \\
&= (I - \mathbf{1}P_2^\top)A - (I - \mathbf{1}P_2^\top)AP_1\mathbf{1}^\top \\
&= (I - \mathbf{1}P_2^\top)(A - AP_1\mathbf{1}^\top) \\
&= (I - \mathbf{1}P_2^\top)A(I - P_1\mathbf{1}^\top)
\end{aligned}
$$

So:

$$\nabla_{\theta_1}\nabla_{\theta_2}v_2 = T \odot (I - \mathbf{1}P_2^\top)A(I - P_1\mathbf{1}^\top)$$

$\square$

# G   Stag Hunt

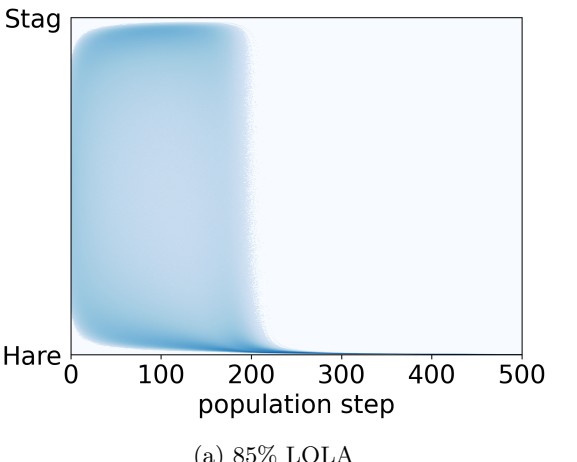

(a) 85% LOLA

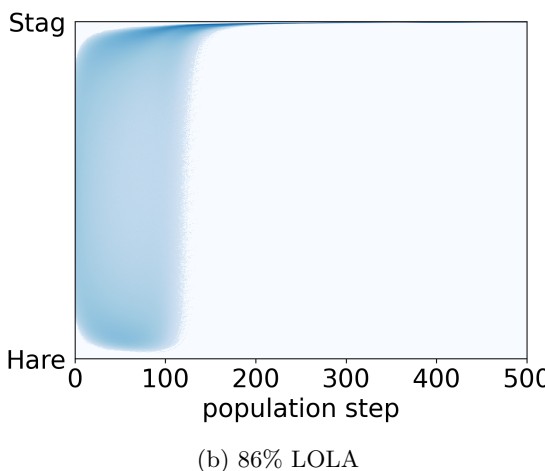

(b) 86% LOLA

Figure 7: Mixed PG and LOLA in Stag Hunt ($s = 1.8$). When more than 86% of the population is made of LOLA agents, opponent-aware learners bring the entire population to the pro-social equilibrium (NB: the higher $s$ is, the lower this threshold becomes; it reaches 0% when $s = 2$).

# H   Hawk Dove

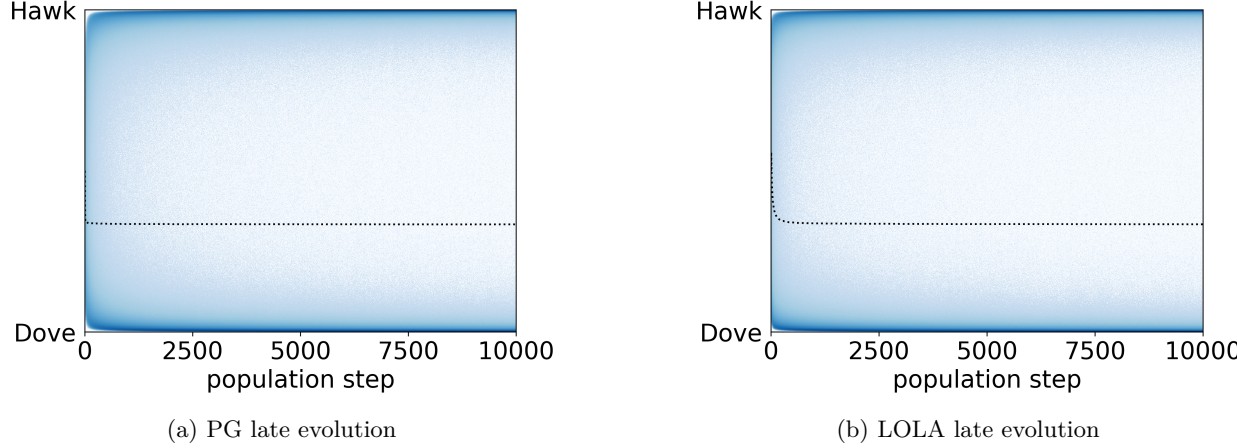

(a) PG late evolution                              (b) LOLA late evolution

Figure 8: Late evolution in Hawk-Dove ($f = -2$)

# I Rock-Paper-Scissors

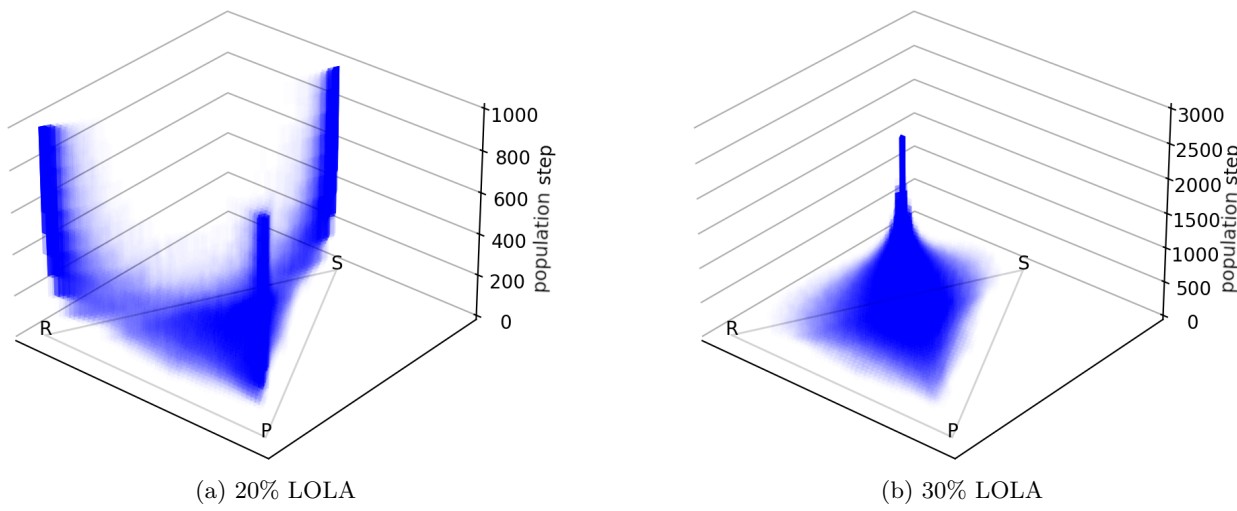

(a) 20% LOLA

(b) 30% LOLA

Figure 9: Mixed PG and LOLA in Rock-Paper-Scissors

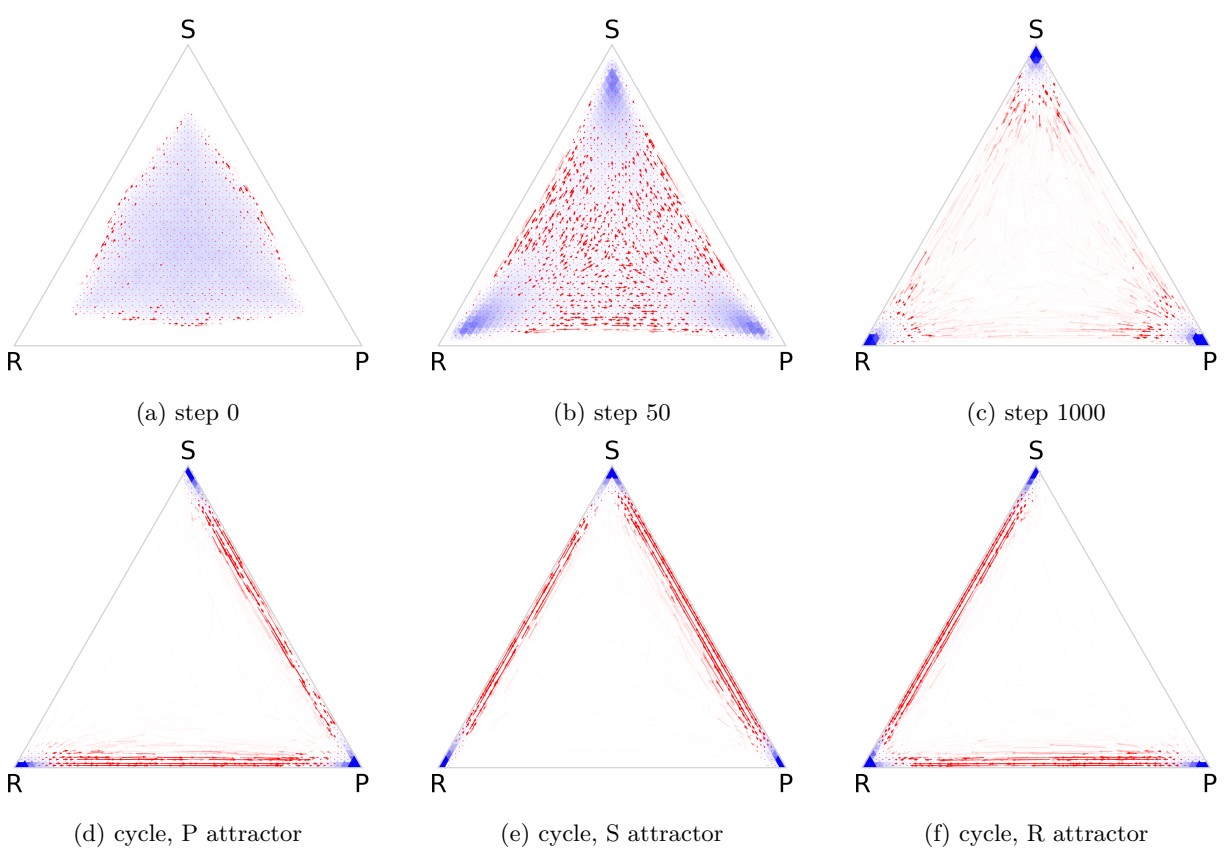

(a) step 0

(b) step 50

(c) step 1000

(d) cycle, P attractor

(e) cycle, S attractor

(f) cycle, R attractor

Figure 10: Naive learning in RPS, late evolution. Shades of blue indicate the concentration of individuals, while red arrows indicate their average measured movement. After about 4000 population steps, random drift slightly unbalances the 3 groups of near-deterministic individuals, which generates a cyclic attractor pattern in the population. In the RPS model, naive learning sustains diversity.

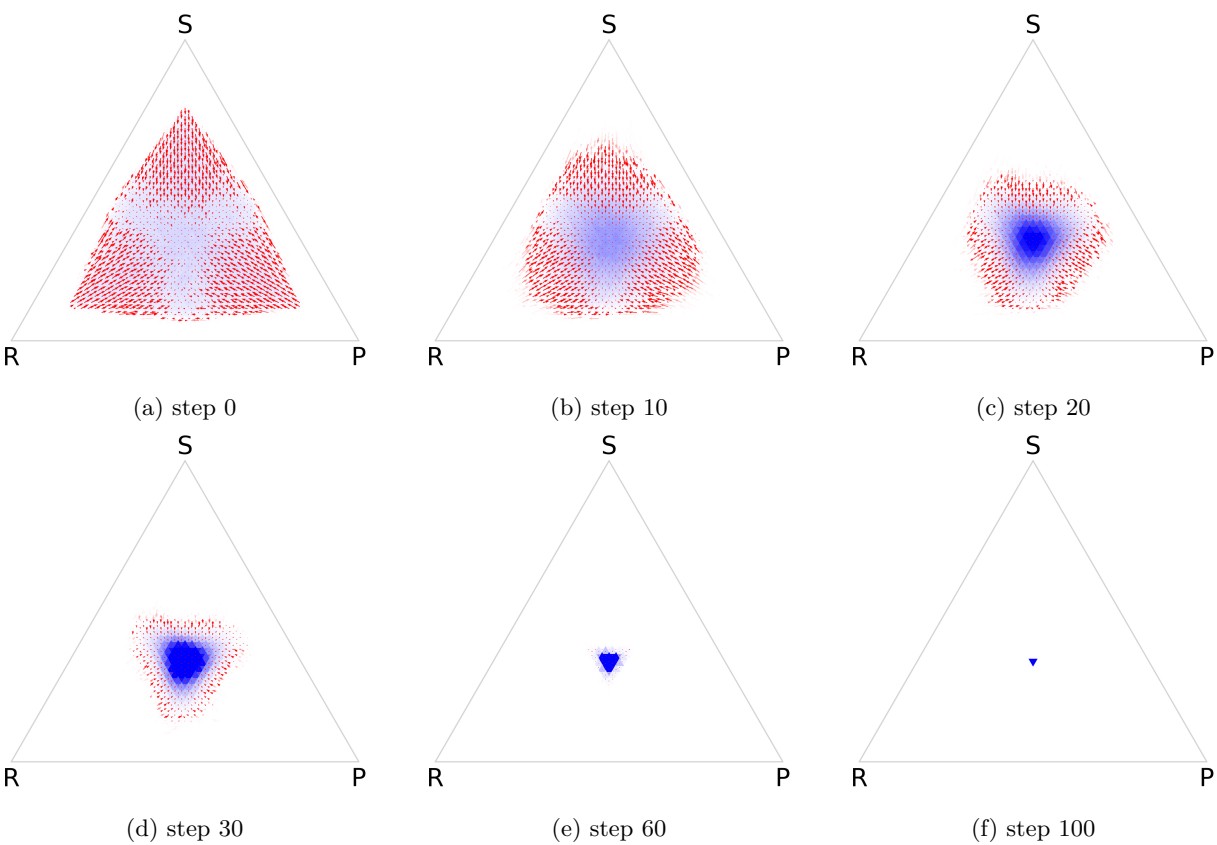

Figure 11: LOLA in RPS. Opponent-learning-awareness quickly brings the entire population to unanimously play the Nash equilibrium of this game (even when 70% of the population is naive, as shown in Figure 9b). In the RPS model, LOLA hinders diversity.

## J  Algorithmic complexity

As noted in Section 5.1, the scalability of the proposed methodology stems mostly from batching operations and leveraging the parallel computation capabilities of machine learning frameworks. In terms of "simple" algorithmic complexity (i.e., ignoring parallelization and framework optimizations for, e.g., tensor multiplications and gradient computations), all approaches tested in Figure 2 are $O(NA^3)$, where $N$ is the number of individuals in the population and $A$ is the number of actions in the game. When using views, the procedure described in Section 4.4 is $O(N)$ because of shuffling, and the computation of each term in Equation 12 is $O(NA^3)$. This is because $P_1$ and $P_2$ are of size $N \times A$ when batched across all $N$ individuals, and considering the schoolbook algorithm of matrix multiplications. As for automatic differentiation, the backward procedure is $O(A^3)$ per individual in the population, i.e., $O(NA^3)$. More details are available in our code.

