# OpenReview forum: "Sociodynamics of Reinforcement Learning"
_TMLR — Accepted by TMLR_

### Review · Reviewer_sMC6 · 2025-10-15

**Summary Of Contributions:**

The authors investigate an efficient implementation of a policy gradient (PG) algorithm that incorporates opponent-aware learning considering large population game. The main contribution lies in the proposed matrix-batched method, which randomly pairs agents that enables scaling to large population game. The claims are mainly supported by experimental results, including Stag Hunt, Hawk-Dove and RSP. The opponent-aware learning method shows different behavior form original policy gradient approach and the differences depend on the game class and whether training is in self-play.


**Strength**

1. The paper efficiently observes distinct behavioral patterns for opponent-aware learning versus  naive PG algorithm.
2. It is empirically shown that the proposed method scales to large population dynamics of PG thanks to efficient computation of matrix calculation.
3. The paper is well-written and provides sufficient background to understand the paper.

**Weakness**

1. Even though the authors compare the proposed algorithm in several well-known examples, it is unclear why such difference happens and how we should understand it. It seems that the behavior vary by game class and by whether training is in self-play. What is the main take-away from these experiments?

2. The paper lacks theoretical analysis on the dynamics of LOLA in (12). For example, what are the conditions of its convergence and how fast does it converge?

3. It is not clear how efficient the method is compared  to original method of the computation comes from. Qualitative analysis of the computational complexity could be provided to clarify this point.

**Audience:**

Yes

**Audience Explanation:**

The paper deals a problem of learning dynamics of evolutionary dynamics in particular, the policy gradient algorithm. Empirical results on standard benchmarks further underscore its relevance to both the game-theory and reinforcement-learning communities

**Claims And Evidence:**

No

**Claims Explanation:**

The claims are mainly supported by the experimental results. On the other hand, it lacks theoretical justification, e.g., simple computation complexity analysis on matrix batched method or convergence (rate) of proposed LOLA algorithm.

**Requested Changes:**

1. As mentioned in the weakness, theoretical analysis of the convegence or convergence rate of the update of LOLA in (12) could strengthen the paper.


2. Providing analysis of computational complexity of the proposed method (matrix batched method) could strength the understanding of the proposed method.

3. Could the authors clarify whether Figures 3(a) and 3(d) are the same plot? They appear indistinguishable.

---

> ### Author Response · Authors · 2025-10-28
> **Added the requested computational complexity analysis and clarifications of Figure 3**
>
> We thank you for your insightful comments and for the time spent reviewing our manuscript.
>
> Because you found that our work is of interest to the TMLR audience but lacks theory to back our claims, we hope to address this concern in priority.
>
> Our paper is a first step toward understanding the dynamics of population games played by RL-enabled learners. As you have rightly noted, it does mainly revolve around simulation and empirical results.
>
> We agree that a sound theoretical analysis explaining our simulation results would be great, and this is one long-term objective of our work. However, MARL-driven population dynamics are much harder to analyze than the replication-driven dynamics typically studied in EGT. Thus, the answer to your question in weakness (2) and requested change (1) is that, at the moment, we do not know how to derive theory that would explain why a large population of agents following the LOLA algorithm (derived for population games in Equation 12), or even simply following the vanilla PG algorithm (similarly in Equation 10) converges at a given speed in a given population game, or whether it converges at all (see Hawk-Dove).
>
> The ‘self-play’ analysis presented in Figure 3 is the closest thing we have to a theoretical predictor of these dynamics, as we explain in the first paragraph of Section 5.2. Studying the convergence properties of LOLA in simple self-play seems possible and has been touched upon in [1]. But a theoretical analysis of self-play would still clearly not be a sufficient predictor of population dynamics: for instance, in Hawk-Dove, PG and LOLA converge to two different policies in self-play (Fig. 3 b and e), whereas populations of either PG or LOLA agents do not converge to any specific policy. Rather, they converge to a dynamic mix of policies averaging to the PG self-play policy in both cases, even when the population is purely made of LOLA agents (Fig 4 b and e, Fig 5 b). This is one takeaway from our experiments, to answer your question in weakness (1) (we would also like to make sure you understand why we conduct those self-play experiments, which are 2-cloned-agent experiments, not population experiments). Other takeaways are mainly the simulation results presented in Figure 4, which describe how populations of either “naive” (PG) or non-stationarity-aware (LOLA) learners evolve in three fundamental interaction models. We find these results insightful and believe they may have an impact on the future of social sciences. They are what we consider the main point of our paper.
>
> You have suggested that we provide complexity analysis for the performance of the matrix-batched approach (Requested change 2). Please note that Figure 2 is already an empirical proxy for this complexity analysis. Also, keep in mind that the scalability of both our batched approaches resides principally in making the problem embarrassingly parallel and leveraging deep learning frameworks’ GPU optimizations for parallel matrix calculus and gradient computations. Nevertheless, since the requested analysis is easy, we have added it to our Supplementary Material (code) to answer your comment. The theoretical complexities of the matrix-based approach and of the autograd-based approach differ only by a constant  that explains the superiority of the matrix approach seen in Fig. 2. Both the matrix-based and the autograd-based approaches have a complexity of order O(NA³), where N is the population size and A is the number of actions. This is true for both PG and LOLA, but LOLA has more sequential matrix operations and thus is more computational by order of another constant.
>
> Requested change 3: Figure 3(a) and 3(d) are not the same plot and should definitely not appear indistinguishable: we have added colored lines to these two plots in order to make the difference stand out more in response to your question. The difference resides in where the fork is located with respect to the grey dotted line, which represents the neutral policy (choosing Stag or Hare uniformly at random). The takeaway from this plot is that, in single-agent self-play, a “naive” PG agent converges to the non-cooperative strategy (Hare) unless its initial policy is already strongly biased toward the cooperative strategy (Stag), whereas a self-play LOLA agent converges to the cooperative strategy even if its initial strategy is slightly biased toward the non-cooperative strategy. This plot is to be considered in conjunction with Figure 5(a), which shows a similar effect in actual populations of PG and LOLA agents (see also plots 4(a) and 4(d)). Please also note the following sentence in the text of Section 5.2: “Notice that the forks are on either side of the uniform random policy (middle tick) [...]”.
>
> (IMPORTANT: one review is still missing; we will upload our revised submission when it arrives, as per guidelines.)
>
> References:
>
> [1] Letcher, A., Foerster, J., Balduzzi, et al. (2018). Stable opponent shaping in differentiable games

---

> ### Author Response · Authors · 2025-12-21
> **Computational analysis added in the supplementary material**
>
> We have also included the Jupyter notebook (.ipynb) shared earlier, provided as a zipped supplementary file. This notebook contains the implementation along with detailed comments, including the aforementioned complexity analysis.

---

> > ### Comment · Reviewer_sMC6 · 2025-12-28
> >
> > Thanks the authors for the detailed response. I have still a few questions on the computational complexity of the proposed method :
> >
> > 1. Why does the the matrix-based and the autograd-based approaches have computation complexity by O(NA^3)? It the key advantage from parallel computation?
> >
> > 2. Moreover, I could not locate where the authors provided computational complexity in their supplementary code.

---

> > > ### Author Response · Authors · 2025-12-30
> > > **Precisions on algorithmic complexity**
> > >
> > > Hello,
> > >
> > > The matrix-based approach has an algorithmic complexity of O(NA^3) (upper bound) for both PG and LOLA because, when we batch the population together, the P_1 and P_2 vectors in Equation 12 become of size (N x A) and considering the naive matrix multiplication complexity of O(MN^2) (schoolbook algorithm). So, for instance, computing the term P_1 \odot X_1 A P_2 involves a series of matrix operations whose maximum complexity is O(NA^3).
> > >
> > > Regarding the autograd-based approach, each gradient computation is of complexity O(A^3) and this is repeated across the entire population, i.e., N times.
> > >
> > > We have added a small section at the end of the Appendix to clarify these points.
> > >
> > > The detailed complexity analysis is provided line-by-line in the comments of our implementation of those approaches in the Jupyter notebook downloadable as a .zip file on OpenReview under "Supplementary material". The relevant cells are called "Matrix-based simulation (batched update)" and "Gradient-based simulation (batched update)".

---

### Review · Reviewer_o5Te · 2025-10-15

**Summary Of Contributions:**

The authors state the main contribution of the paper is a methodology for using reinforcement learning to learn strategies that reach stable equilibria in multi-agent settings where all agents seek to maximize their individual rewards.

**Weaknesses**
- This paper was very difficult to read and I had a hard time going. Consequently, neither the novelty nor the significance of the paper's contributions are readily apparent. Maybe I missed something or maybe I am not the right audience for the paper, but I did not come away with any clear message.
- The authors' justification for using reinforcement learning to learn stable equilibria strategies when considering the evolution of population dynamics i.e., the EGT framework, is tenuous. The strongest justification being that their is some work that show organisms use some degree of reinforcement learning to learn behavior.
- I also do not understand why the authors choose to fit policy gradient and LOLA (I get why LOLA would be used over policy gradient, but why apply it to the EGT framework?) to EGT. Approaches like *regret matching* and *fictitious self-play* seem to be obvious for the problem. They also iteratively learn policies through repeated interactions so can one not argue that they are also RL approaches? They also seem to be more suitable for EGT. If not, why not? Why use PG/LOLA over them? What do we gain or learn by applying PG/LOLA instead?
- The paper is also not well-written. It was difficult to understand most of what was written. Some examples of problems:
    - The subsection on EGT never clearly explains what the goal is.
    - Many sentences are really long and hard to follow.
    - The authors do not explain the purpose of experiments or what they intend to show with the results.

**Audience:**

No

**Audience Explanation:**

The paper offers no significant contributions, in my opinion.

**Broader Impact Concerns:**

No concerns.

**Claims And Evidence:**

No

**Claims Explanation:**

It is hard to identify what the claims actually are or which claims are novel to the paper.

1) They claim to introduce a way to implement policy gradient and LOLA in a backprop-free manner that can be parallelized.
    - They say Eq 7 is what allows them to do this for PG, for instance. Is that novel? If it is, then the authors need to prove or derive the equation. They simply state that this can be done. Is it because it has been shown in the literature? Then it isn't a contribution.
2) I don't quite know what claim(s) the experiments are trying to support. They compare an approach that treats the entire population as a single agent and their approach that treats the population as multiple agents. The message from the results that I took away was that treating the population as a set of agents is better. Yes, that seems like an easy conclusion to come to given the nature of the problem. What new insight do I get from these results?

**Requested Changes:**

I do not think there are any changes that would make me change my mind on the paper. The paper needs to be re-written so that its significance is more explicitly stated and justified, and their contributions are clearer.

---

> ### Author Response · Authors · 2025-12-21
> **Clarification of the main contribution of the paper.**
>
> We thank you for the time reviewing our manuscript.
>
> We are sorry to find that you were likely not the intended audience of our paper and thus could not understand our contribution: we are not particularly interested in stable equilibria, Equation 7 is a trivial and known result that is not meant as a contribution, PG and LOLA are classics in Multi-Agent Reinforcement Learning theory (including in our setting, see [1,2,3]), etc.
>
> Our paper leverages simulation to bring elements of answer to the question “what happens when a large population is made of individuals who all actively seek to optimize their own incentives, either through naive (PG) or non-stationarity-aware (LOLA) multi-agent reinforcement learning?”. This work is principally relevant for modeling economy and social dynamics, and it might also have an impact in biology [4].
>
> Since the topic is complex and theoretically hard, we proposed a simplified approach based on EGT-like modeling and simulation to obtain the elements of answer summarized in Figure 4. We consider Figure 4 to be our main contribution. Previous Sections describe our methodology, which enabled us to conduct the otherwise intractable simulations presented in Figure 4. Understanding the mathematical derivation of this methodology is more involved (see Supplementary Material), but is not required to understand our main takeaways.
>
> To be clear, we are not interested in “learning strategies that reach stable [Nash] equilibria”. We are instead interested in modeling how large populations made of learning individuals behave.
>
> Since the two other reviewers have found the paper well-written and easy to follow, we do not intend to go through a major revision for clarity. Nevertheless, we are happy to clarify any specific sentence that sounds confusing in the manuscript, or to implement specific suggestions that would make the contribution clearer to a broader audience.
>
> References:
>
> [1] Letcher, A., Foerster, J., Balduzzi, D., Rocktäschel, T., & Whiteson, S. (2018). Stable opponent shaping in differentiable games. arXiv preprint arXiv:1811.08469.
>
> [2]: Yang, Y., Luo, R., Li, M., Zhou, M., Zhang, W., & Wang, J. (2018, July). Mean field multi-agent reinforcement learning. In International conference on machine learning (pp. 5571-5580). PMLR
>
> [3]: Bloembergen, D., Tuyls, K., Hennes, D., & Kaisers, M. (2015). Evolutionary dynamics of multi-agent learning: A survey. Journal of Artificial Intelligence Research, 53, 659-697.
>
> [4]: Macy, M. W., & Flache, A. (2002). Learning dynamics in social dilemmas. Proceedings of the National Academy of Sciences, 99(suppl_3), 7229-7236.

---

### Review · Reviewer_N5h7 · 2025-12-11

**Summary Of Contributions:**

This paper proposes Policy Gradient (PG) and Opponent-Learning Awareness (LOLA) based techniques to study the learning behaviors of large populations of agents. Experiments on three stateless matrix games with populations of 200K agents demonstrate the efficacy of the proposed algorithm. This is an interesting contribution that aims to push the boundaries of multi agent reinforcement learning (MARL) to large populations of learners.

Strengths:
•	The paper presents fast, vectorized implementations of PG and LOLA updates that scale to populations of 200,000 agents, which is technically impressive.
•	Authors derive update rules for PG and LOLA, and provide mathematical approach for large-scale simulations.
•	The experiments aim to illustrate how LOLA and PG learners influence cooperation and diversity in classic games (Stag Hunt, Hawk-Dove, RPS), which is conceptually interesting (promoting or delaying the cooperation, and reducing the strategy diversity).

Weaknesses:
•	Unsupported claim on heterogeneity
•	Mathematical notation issues
•	Limited baselines

**Additional Comments:**

The overall direction of the work is promising, and the connection between EGT and scalable MARL is valuable. Strengthening the empirical evaluation and clarifying several presentation issues would substantially improve the clarity and impact of the paper.

**Audience:**

Yes

**Audience Explanation:**

The intersection of evolutionary game theory, multi-agent reinforcement learning, and large-scale population simulations is of clear interest to the TMLR community. The paper attempts to connect well-established MARL ideas (PG, LOLA) with scalable population-level models, which is relevant for both theory and applications (e.g., collective behavior, societal modeling).

**Broader Impact Concerns:**

I do not see major ethical concerns beyond standard considerations for research on learning dynamics and social simulations. The work does not appear to require a broader impact statement beyond what is already typical for MARL and game-theoretic modeling.

**Claims And Evidence:**

No

**Claims Explanation:**

•	The abstract claims that the proposed algorithm can simulate heterogeneous agents. However, none of the experiments actually demonstrate heterogeneity. If the authors wish to support this claim, they should define what they mean by heterogeneity (e.g., learning rules, initialization distributions, policy parameters, payoff matrices, or population structures) and include corresponding empirical results.

•	The introduction cites Foerster (2018) as the primary reference for MARL, but MARL is a much older field dating back at least to Tan 1993 [Tan, M., 1993. Multi-agent reinforcement learning: Independent vs. cooperative agents. In Proceedings of the tenth international conference on machine learning (pp. 330-337).].

•	In section 3.1, 3.2 PG, LOLA, and first-order Taylor expansion are mentioned without citation. Introduction of Taylor expansions also lacks a clear mathematical citation.

•	Parameter “P” is defined as both population vector and probability vector which makes the notation confusing.

•	The proposed method is benchmarked against only two baselines. The results presented for “autograd iterative LOLA” baseline appear to diverge or explode visually, making them incomparable to the proposed method. This raises concerns about both baseline selection and the clarity of presentation.

•	The captions for the figures 1, 2, 3, 5, 7, 8 are very short and not informative for readers to interpret them without referring to the main text.

•	Caption for figure 3 mentions that “the color marks the initial policy but the specific color mapping is not explained.

•	There are many interesting similar works in the literature for mean field games (see for instance Hu et. Al 2025 https://link.springer.com/article/10.1007/s00245-025-10328-5) which are more recent and can serve as meaningful baselines. However, the most recent citation in the paper is from 2022.

•	In section 5.3 authors said “From our results, it looks like …” is overly informal and introduces doubt.

•	It is not clear if the results are presented for more than 1 replicate which is crucial for probabilistic models.

•	Figures 3, 4, 6, 7, 8, 9, 10 requires a color bar or heatmap for explanation.

•	It is mentioned that the learning rate does not impact the algorithm , but no ablation experiments are provided to support this claim.

**Requested Changes:**

Critical changes (required for acceptance):

The claim of supporting heterogeneous agents must be demonstrated experimentally or removed; authors must clearly define what “heterogeneity” means (learning rules, initialization, payoff differences, etc.).

Add foundational MARL citations (e.g., Tan 1993) to correct the misleading historical context.

Add proper citations for PG, LOLA, and first-order Taylor expansions in Sections 3.1 and 3.2.

Resolve the ambiguous notation where P refers to both population vectors and policy probability vectors.

Improve the baseline evaluation: the autograd LOLA baseline diverges visually and is not informative; include more meaningful baselines, especially recent mean-field MARL methods.

Provide multiple replicates or statistical reporting, as single runs are insufficient for stochastic population dynamics.

Improve figure clarity: add descriptive captions and include colorbars/heatmaps for Figures 3, 4, 6, 7, 8, 9, and 10.

Provide ablation results supporting the claim that learning rate has no effect.


Recommended (non-critical) improvements:

Clarify color coding in Figure 3 (“the color marks the initial policy”) with explicit mappings.

Update the related work section to include more recent literature (post-2022).

Replace informal phrasing such as “From our results, it looks like …” in Section 5.3 with precise statements.

Restructure figure presentation to avoid extremely large or diverging baselines that distort comparisons.

---

> ### Author Response · Authors · 2025-12-17
> **Addressing minor comments before christmas Part 1. (Detailed response to follow)**
>
> We appreciate your careful and diligent review. We agree with most of the points raised in your comments, and we wish to address them properly.
>
> Since the rebuttal period is short and near the Christmas season, we have released a first revision to address the points that we believe are easiest to address, and to engage the discussion. We will later release a more complete revision to address the remaining points.
>
> $\textbf{Claims on heterogeneity}$: The abstract stated that we simulate large populations of “heterogeneous agents”. What we meant here was just that these agents are not self-play agents and instead have independent policies. Nevertheless, you are right to note that heterogeneity in our work extends beyond this: the fact that agents have heterogeneous (i.e., non-self-play) policies does yield heterogeneous population structures (e.g., Figure 4.c.), and in the supplementary material, we do conduct experiments where agents have heterogeneous learning rules (Sections G, I). Furthermore, with our approach, it would be straightforward to conduct more experiments on heterogeneity. For example, we could easily explore initial distributions beyond simple gaussians. However, in the simple games that we selected, this would not yield particularly interesting results: the observed heterogeneity in population structure in our current work is a combined consequence of random drift and of loss of plasticity, not of initial distribution.
>
> - Change: for now, we have simply replaced the word “heterogeneous” with “independent”, although we may reintroduce it later, depending on this discussion.
> - Question: Can you please clarify what you mean by “actually demonstrate heterogeneity”?
>
> $\textbf{Foerster as primary reference for MARL}$: Thank you for the historical reference.
> - Change: we have added Tan, M., 1993.
>
> $\textbf{Missing citations in Section 3.1 and 3.2}$:
> - Change: the requested citations for EGT (3.1) and PG/LOLA/Taylor series (3.2) have been added.
>
> $\textbf{Parameter “P”}$: the reason behind our choice of identical letters for population vectors and policies is described in Section 3.3. This equivalence motivates our comparison between populations of RL agents and single pairs of self-play agents in Section 5.2.-
> - Change: we have underlined all symbols when they primarily refer to population vectors (Equations 1 and 2).
>
> $\textbf{Baselines in Figure 2}$: One specificity of our paper is that we are not interested in finding RL algorithms. Instead, we are interested in studying how large populations evolve when they are made of simple, classic self-interested MARL agents, either naive or non-stationarity-aware (PG or LOLA). Thus, our “baselines” in Figure 2 only compare our own implementations of LOLA together, purely in terms of wall-clock time, as this is crucial for our full large-scale simulations. The iterative implementation does explode because it is not taking advantage of the simple parallelization tricks described in Section 4.4. In terms of RL and population dynamics, all 3 implementations do the same thing: analytical LOLA.
> - Question: It is unclear to us how to compare our approach to other baselines in Figure 2. As far as we understand, this would likely amount to comparing the population dynamics yielded by PG and LOLA to that of other algorithms, which is not the point of Figure 2 and is beyond the scope of our paper.

---

> ### Author Response · Authors · 2025-12-17
> **Addressing minor comments before christmas Part 2. (Detailed response to follow)**
>
> $\textbf{Regarding mean-field approaches as baselines}$: While mean-field methods are indeed relevant to large-population dynamics, they address a fundamentally different question than our work. Mean-field approaches (e.g., Yang et al. 2018, Hu et. Al 2025) reduce the complexity of multi-agent interactions by approximating neighboring agents as a single "mean-field" opponent, effectively transforming n-agent MARL into pairwise MARL. This is a simplification technique that reduces environment dimensionality from an individual agent's perspective. In contrast, our work explicitly tracks the full heterogeneous distribution of 200,000 independent policies precisely because we are interested in emergent population-level phenomena that mean-field approximations would obscure. For instance, in Rock-Paper-Scissors (Figure 4c), naive learners evolve into three distinct clusters of near-deterministic agents - a phenomenon that would be invisible under mean-field assumptions, which would predict convergence toward the population average. This distinction is analogous to recent work on symmetry-breaking bifurcations (Hendriks et al. 2025, "Equivariant Flow Matching for Symmetry-Breaking Bifurcation Problems"), which demonstrates that deterministic approaches fundamentally cannot capture multimodal solution distributions arising from bifurcations. Similarly, mean-field approximations would predict that all agents converge to similar policies, missing the rich population structure we observe.
>  - Change: We will nonetheless add more references to the mean-field literature in our related work section in the next revision to clarify this distinction.
>
>
>
> $\textbf{Short captions}$: The choice of title-style captions in the main paper was guided by the recommended number of pages.
> - Change: Since TMLR does not really enforce space constraints, we have replaced these captions with more self-contained versions.
>
> $\textbf{Color mapping of Figure 3}$:
> - Change: The color mapping is now described in the caption of Figure 3.
>
> $\textbf{Overly informal statement in Section 5.3}$: You are right that this statement should not introduce doubt: its point is to clear a possible confusion
> - Change: replaced with the following formulation: “Our simulations can mislead the reader into concluding that [...]”
>
> $\textbf{Multiple replicates}$: Figure 3 describes deterministic dynamics, and Figure 4 describes near-deterministic dynamics: the large number of stochastically learning agents (200k) averages out to near-deterministic population dynamics. We ran these experiments many times with random seeds and always observed the same results. We can report the variance of the mean policy over several runs if you feel this is relevant, but this variance is virtually 0.
> - Change: for now, we have added the following statement in section 5.3: “Due to the large number of simulated agents, the reported population dynamics are near-deterministic.”
>
> $\textbf{Ablation study on learning rate}$: will be added to the next revision.

---

> ### Author Response · Authors · 2025-12-21
> **Ablation study on learning rate**
>
> In the revised manuscript, we have added a new subsection, “Influence of Learning Rates,” within the empirical results section to analyze the effect of learning rate in our simulations.
>
> Our former quick testing had led us to believe that learning rates only had trivial effects on population dynamics that we did not consider worth mentioning in the paper (namely, that the first-order learning rate scales the speed of the population dynamic and that the second-order learning rate scales the effect of opponent-learning awareness). However, a more in-depth study reveals a surprising phenomenon in the Stag Hunt (SH) interaction model.
>
> In SH, self-play behaves as expected: the first-order learning rate has no effect other than scaling the speed of convergence to either strategy, and the second-order learning rate scales the effect of LOLA until it becomes insufficient to pull the agent toward the cooperative strategy. In stark contrast, large populations of LOLA agents do surprisingly exhibit a combined effect of the first and second-order learning rate on the convergence of the population to either strategy. The ratio between the 1st-order (PG) and 2nd-order (opponent-awareness) terms in LOLA determines whether the population converges to the cooperative (Stag) or individualistic (Hare) equilibrium. This effect is particularly relevant to our study since it seemingly emerges from the population setting. However, at the moment, we do not know why this phenomenon emerges (especially since it does not happen in self-play), and thus we simply showcase it as an interesting curiosity in Section 5.3.

---

### Author Response · Authors · 2025-12-24
**Summary of changes in the revised manuscript**

Dear reviewers and action editor,

We have published a revised version of our manuscript and supplementary material following the reviewers’ recommendations and requests. Please find thereafter a summary of the affected items:
- Figure 3(a) and 3(d): added coloured dotted lines on critical policies where the bifurcation happens in self-play (requested by sMC6)
- Supplementary material: added computational complexity analysis in the comment (requested by sMC6)
- Temporarily removed the term “heterogeneous” from the abstract (requested by N5h7)
- Added relevant references in Section 1, 2 and 3, including more recent references on mean-field MARL (requested by N5h7)
- Adapted our choice of notation in equations 1 and 2 to avoid confusion around P when it primarily refers to a population vector rather than an RL policy (requested by N5h7)
- Changed the captions of all figures in the paper to make them more descriptive such that all figures are more self-contained (requested by N5h7)
- Clarified a confusing statement in section 5.3 (requested by N5h7)
- Explicitly stated in Section 5.3 why multiple replicates and statistical confidence bounds are unnecessary in our work (in response to N5h7)
- Added a new section exploring the influence of learning rates (Section 5.3). In particular, this section uncovers a surprising effect of both learning rates (1st and 2nd order of LOLA) in the Stag Hunt interaction model when played by large populations (requested by N5h7)

---

### Comment · Action_Editor_iVyQ · 2026-02-11
**Camera ready version**

Looking at your camera-ready version, it seems to me that you used \clearpage or \newpage after the abstract, so the Introduction starts on a new page. That does not adhere to TMLR's formatting guidelines. Can you please fix this issue?

---

> ### Author Response · Authors · 2026-02-12
> **Compliant formatting**
>
> Hello, I had indeed added a \newpage, sorry. I removed it in the updated version to comply with the TMLR guidelines and I did some minor rephrasing in the introduction to optimize the layout.

---

> > ### Comment · Action_Editor_iVyQ · 2026-02-19
> >
> > Hello,
> >
> > I'm sorry for the back-and-forth, but as I look at the new version, is there a reason Figure 3 is on its own page? I imagine that if you anchor it at the top of the page, a substantial amount of text from what is now page 9 would go on that page. This is a genuine question: I'm trying to care for the look and feel of the paper, as it looks weird with pages 7 and 8 being only figures. It is unclear that there's more to come.
> >
> > I appreaciate your patience,
> >
> > Cheers.

---

> > > ### Author Response · Authors · 2026-02-21
> > > **Separated figures**
> > >
> > > Hello,
> > >
> > > There was no particular reason for this choice, it was how Latex chose to organize Figures 3, 4 and 5 I believe, but it is true that this layout looked weird. We have posted a revised version where we reorganized how these figures appear, such that they don't break the flow of text anymore.

---

### Decision · Action_Editor_iVyQ · 2026-01-13

**Recommendation:** Accept with minor revision

**Additional Comments:**

See meta-review above w.r.t. the requests made by reviewer N5h7.

**Audience:**

Yes

**Audience Explanation:**

This is a paper that reviewers agree the MARL community might be interested in.

**Claims And Evidence:**

Yes

**Claims Explanation:**

This paper empirically investigates the behaviour of a population of agents who optimize their own individual rewards via reinforcement learning. It investigates both policy gradient and non-stationarity-aware methods in well-known games: Stag-Hunt, Hawk-Dove, and RSP. A key contribution is how to address scaling issues, achieved via pairwise matching and matrix-batched updates.

I am recommending the paper’s acceptance. During the review and discussion phase, concerns were raised about the presentation of the paper—including concerns around making some of its terminology more precise, references more accurate—, and the lack of a theoretical analysis in the paper. The authors addressing some of the presentation issues was important for this recommendation, while I do agree with the authors that the theoretical analysis should’t be a requirement for such an investigation be published. That said, the minor revisions I am requesting still revolve around the issues raised by reviewer N5h7, with the most prominent example being the use of the word 'heterogeneity'. The authors acknowledged that they sent only a preliminary draft of the final version of the paper due to timing issues.

I want to apologize to the authors for the long delay in reaching a final decision on this paper, as I struggled more than usual to find reviewers and had to replace some during the process. In any case, at least these are good news, and I get to congratulate you on the paper's acceptance.

---

> ### Author Response · Authors · 2026-02-09
> **Camera-ready version**
>
> Thank you for recommending our work.
>
> Since reviewer N5h7 did not respond to our questions regarding heterogeneity, we have decided to replace all mentions of "heterogeneous" with "independent" in our camera-ready version to avoid misleading expectations when, e.g., reading the abstract (note: "heterogeneous" referred to learning processes and to final learnt policies / population structures in our paper).
>
> We have also optimized the layout and finished implementing all of the reviewers' suggestions that we agree on.
>
> We again thank all reviewers and meta-reviewer for their useful suggestions and for the time reviewing this paper.